# Do better language models have crisper vision?

## Abstract

How well do text-only Large Language Models (LLMs) grasp the visual world? As LLMs are increasingly used in computer vision, addressing this question becomes both fundamental and pertinent. However, existing studies have primarily focused on limited scenarios, such as their ability to generate visual content or cluster multimodal data. To this end, we propose the Visual Text Representation Benchmark (ViTeRB) to isolate key properties that make language models well-aligned with the visual world. With this, we identify large-scale decoder-based LLMs as ideal candidates for representing text in vision-centric contexts, counter to the current practice of utilizing text encoders. Building on these findings, we propose *ShareLock*, an ultra-lightweight CLIP-like model. By leveraging pre-computable frozen features from strong vision and language models, *ShareLock* achieves an impressive $51\%$ accuracy on ImageNet despite utilizing just 563k image-caption pairs. Moreover, training requires only 1 GPU hour (or 10 hours including the precomputation of features) – orders of magnitude less than prior methods. Code will be released.

## 1 Introduction

Large Language Models (LLMs) are solely pretrained on unimodal textual data, yet they are increasingly incorporated into systems that perceive and interact with the natural world (Ahn et al., 2022; Driess et al., 2023; Wayve, 2023). The lack of direct sensory experience raises fundamental questions to which extent such models can develop a meaningful and accurate understanding of *visual* reality. Do these models merely regurgitate visually relevant factual knowledge from their training corpus, or do they form internal representations that correspond to real-world phenomena? Despite the successful integration of LLMs into large-scale Vision-Language Models (VLMs) (Liu et al. (2023); Li et al. (2023); OpenAI (2023)), it is difficult to judge the visual capabilities already inherent to LLMs this way. This is not only because of the widely varying training recipes and proprietary data sources but particularly due to the fine-tuning with *paired* image-text data, which dilutes and overrides any visual knowledge already contained in the text-only model.

In contrast, Sharma et al. (2024) and Huh et al. (2024) more immediately assess the visual nature of LLMs and highlight a non-trivial degree of visual understanding and cross-modal alignment. These works do this by compiling proxy tasks or measures such as generating code to represent visual concepts (Sharma et al., 2024) or correlating visual with language-based representations (Huh et al., 2024). However, the reliance on highly constrained and synthetic tasks with limited practical significance fails to gauge the aptitude of LLMs when deployed in more realistic settings.

To this end, we propose the Visual Text Representation Benchmark (ViTeRB), a novel benchmark that directly measures performance on the downstream task of zero-shot open-vocabulary image classification, as popularised by CLIP (Radford et al., 2021). This enables us to quantify the visual understanding of language models and their ability to encode text for vision-centric tasks. To prevent concept leakage during the training stage – a significant factor underlying the robust "zero-shot" performance of many VLMs (Fang et al., 2022; Udandarao et al., 2024; Parashar et al., 2024) – we revert to the traditional notion of zero-shot learning (ZSL) where seen and unseen concepts can be strictly delineated and are disjoint (cf. (Lampert et al., 2009)). With the advent of VLMs like CLIP (Radford et al., 2021), these formerly strict assumptions have been watered down in favor of scaling to large volumes of web data that likely contain most but the rarest entities and objects. Consequently, by enforcing a clear training and evaluation protocol, we can accurately assess the *true* generalization capabilities facilitated by the language embeddings.

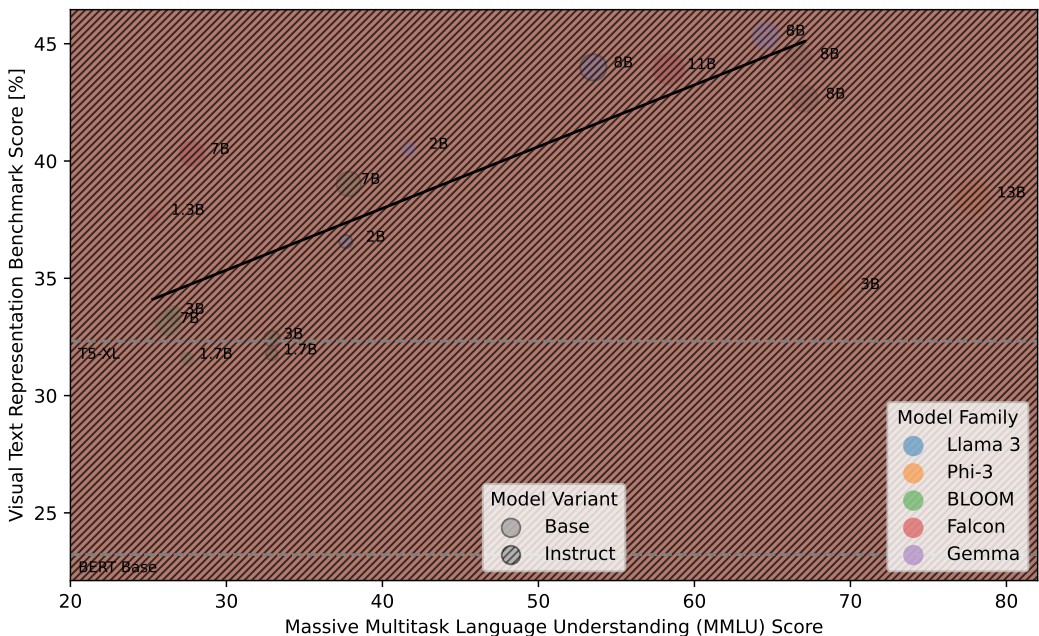

Figure 1: **ViTeRB performance relative to MMLU scores.** Model capability on language tasks is predictive of visual transfer performance as measured on our Visual Text Representation Benchmark ($R^2$: 0.379 and 0.075 (excl./incl. Phi-3 models)).

Using the ViTeRB benchmark, we investigate what properties and design choices enable language models to be effectively leveraged in vision-centric tasks. As one of our key results, we find that features extracted from decoder-based LLMs are more effective compared to encoder-based embeddings. Intriguingly, we find that general LLM capability, as measured through MMLU (Hendrycks et al., 2021), correlates positively with the model's ViTeRB visual performance, as shown in Figure 1. Even off-the-shelf, text-only LLMs without embedding-specific fine-tuning demonstrate strong visual representation abilities.

Based on these findings, we propose "**Share**d Vision-Language-**Lock**ed Tuning" (*ShareLock*), a straightforward late-fusion VLM that leverages the expressive representations of frozen models across both modalities. With vast streams of unimodal data available for large-scale unsupervised pretraining, our research question is how to optimally exploit this resource and investigate how little image-text *paired* data, and thus weak human supervision, is needed to achieve competitive results.

Our extensive evaluation of *ShareLock* demonstrates the effectiveness of our approach in a variety of tasks. *ShareLock* outperforms existing methods trained on the same data, such as CLIP (Radford et al., 2021) or LiT (Zhai et al., 2022), by a significant margin on classification problems and performs competitively on retrieval and compositional reasoning problems. With a fraction of the data and learnable parameters, our method approaches the performance of CLIP models fully optimized on orders of magnitude more data. Moreover, by only training a single MLP on top of frozen representations, *ShareLock* is an extremely lightweight framework that allows us to train our model with batch sizes of 16k on a single A100 GPU.

Summarizing, the main contributions of this work are as follows:

- We introduce ViTeRB, a protocol that strictly controls prior concept exposure, enabling the assessment of true visual zero-shot understanding of language models.

- Our benchmark highlights decoder-based LLMs as effective sources of visual knowledge, with semantically meaningful representations directly extractable from their internal states.

- We propose *ShareLock*, a lightweight method that aligns frozen unimodal features, achieving state-of-the-art data efficiency and superior performance compared to previous models.

## 2 RELATED WORK

**Visual understanding of large language models.** Many previous works (Liu et al., 2023; Wang et al., 2023; Li et al., 2023) enable LLMs to interact with visual information by mapping image features into the token embedding space of the language model, an approach that requires extensive alignment on multi-modal corpora. However, LLMs can also infer and reason about visual content without explicit multi-modal training (Bowman, 2023). By transcribing images into text form using separate VLMs, LLMs can be naturally interfaced via language (Hakimov & Schlangen, 2023). Sharma et al. (2024) tasked LLMs to draw common objects and scenes using simple shapes, indicating present spatial understanding and illustrating that LLMs can conceptualize real-world settings. Various works highlight the plausibility and utility of LLM-generated descriptions of objects in the context of image classification and demonstrate that LLMs possess encyclopedic knowledge about visual characteristics (Pratt et al., 2022; Menon & Vondrick, 2023; Yang et al., 2022; Saha et al., 2024). These capabilities suggest that the extensive pretraining on large volumes of diverse textual data aids the visual understanding of LLMs. Prompted by correlations between semantic representations in the language and vision space, Huh et al. (2024) argue that the embedding spaces of neural networks converge towards a shared representation of reality irrespective of the concrete optimization objectives, model architectures, and data utilized during training. Similarly, we investigate the degree of visual alignment inherent to exclusively language-based representations but assess this in the practically more relevant context of zero-shot image classification and design a rigorous benchmark to measure the true generalization capabilities facilitated by language embeddings.

**Data-efficient CLIP-like models.** Prevailing VLMs heavily rely on large-scale corpora. While the original CLIP (Radford et al., 2021) was trained on 400M image-text pairs, ALIGN (Jia et al.) forwent extensive data cleansing, utilizing a total of 1.8B samples and showing that the noisiness of web-scraped data can be offset through scale. However, more recent work suggests improving the data quality rather than quantity as the more promising alley towards better performance, and a litany of filtration methods has been proposed as a result (Schuhmann et al., 2021; Mahmoud et al., 2024; Joshi et al., 2024; Yu et al., 2023). Such investigations aim at identifying data subsets that effectively facilitate generalization while keeping the training recipes fixed (Gadre et al., 2023). Additionally, advances have been made in the model architecture and training regime to improve data and computational efficiency. It has been shown that even smaller language encoders with notably fewer layers can perform similarly to more expressive language models (Cui et al., 2022). Zhai et al. (2022) leverage representations of pretrained image encoders and only tune the text encoder. ASIF (Norelli et al., 2023) takes this a step further by exploiting pretrained encoders for both vision and language modalities and aligning their representations in a training-free manner with only a few million image-text pairs. Compared to these works, our approach focuses on maximizing the utility of existing unimodal models by aligning them with minimal compute and limited paired data.

## 3 *ShareLock*: SHARED VISION-LANGUAGE-LOCKED TUNING

Inspired by the efficiency and effectiveness of late fusion architectures in CLIP-like models, *ShareLock* comprises two separate encoders for vision and language inputs. The outputs of either encoder $\phi(\cdot)$ are subsequently mapped into a shared $d$-dimensional latent space through a projection $\mathbf{p}(\cdot)$. The latent representation for a given input image $\mathbf{x}_i$ or caption $\mathbf{t}_i$ is therefore computed by $\mathbf{z}_{\text{img}} = \mathbf{p}_{\text{img}}(\phi_{\text{img}}(\mathbf{x}_i)) \in \mathbb{R}^d$ and $\mathbf{z}_{\text{txt}} = \mathbf{p}_{\text{txt}}(\phi_{\text{txt}}(\mathbf{t}_i)) \in \mathbb{R}^d$, respectively. Due to the normalization following the projection, the cosine similarity between two embeddings $\mathbf{z}_i$ and $\mathbf{z}_j$ is given by their dot product (i.e., $\text{sim}(\mathbf{z}_i, \mathbf{z}_j) = \langle \mathbf{z}_i, \mathbf{z}_j \rangle$). During training, the contrastive loss encourages the model to maximize the similarity between embeddings of correct image-caption pairings while decreasing the similarity of non-corresponding pairs. For an image-caption pair $i$ in a batch with $N$ items, it is given by

$$\mathcal{L}(i) = -\log \frac{\exp\left(\text{sim}(\mathbf{z}_{\text{m}}^i, \mathbf{z}_{\text{n}}^i)/\tau\right)}{\sum_{j=1}^{N} \exp(\text{sim}(\mathbf{z}_{\text{m}}^i, \mathbf{z}_{\text{n}}^j)/\tau)}, \tag{1}$$

for both alternated modalities pairings $(m, n) \in \{(\text{txt}, \text{img}), (\text{img}, \text{txt})\}$ and with $\tau$ being a fixed temperature parameter. Given a set of classes $\mathbb{C}$ and their corresponding textual class representations $\boldsymbol{f}(\cdot)$ (e.g., "a photo of a <class name>"), the predicted class $\hat{c}$ for a sample $\mathbf{x}_i$ is obtained via $\hat{c} = \underset{c \in \mathcal{C}}{\arg\max} \langle z_{\text{img}}, \mathbf{p}_{\text{txt}}(\phi_{\text{txt}}(\boldsymbol{f}(c))) \rangle$.

Figure 2: **Model Schematic of Late Fusion VLMs.** Compared to prior works like CLIP and LiT, we propose *ShareLock*, which utilizes frozen pretrained representations for both modalities, allowing extremely efficient training. *ShareLock* also benefits from progress in the LLM domain by representing text with decoder-only LLMs, such as Llama-3.

Deviating from prior works, *ShareLock* leverages frozen pretrained models in both the vision as well as language components, as can be seen in Figure 2. As the alignment of the two modalities is still necessary, only the lightweight projection networks $\mathbf{p}(\cdot)$ are optimized.

## 4    VISUAL TEXT REPRESENTATION BENCHMARK

The objective of our proposed visual alignment benchmark ViTeRB is to assess how language models facilitate generalization to novel concepts. It retains the model architecture and optimization objectives of *ShareLock* but places restrictions on the data akin to traditional ZSL approaches (cf. Lampert et al. (2009)). To rigorously attest to the *true* generalization performance without being affected by concept leakage through supervision with arbitrary image-caption pairs, we split conventional image classification datasets into sets of *seen* classes $\mathbb{S}$ as well as *unseen* classes $\mathbb{U}$, ensuring that $\mathbb{S} \cap \mathbb{U} = \emptyset$. To provide coverage across natural and human artifacts (*e.g.*, aircrafts and animals), coarse and fine-grained categories (*e.g.*, zebra vs. dolphin and fish crow vs. American crow), and different scales ($40 \leq |\mathbb{S}| \leq 1000$), the reported scores are averaged per-class accuracies over $\mathbb{U}$ across four datasets. Namely, AWA2 (Xian et al., 2017), CUB (Wah et al., 2011), FGVCAircraft (Maji et al., 2013), and ImageNet$^+$are selected for their complementary characteristics. ImageNet$^+$defines the ImageNet-1k classes as seen concepts and treats the 500 most populated classes (*i.e.*, highest number of training samples) of ImageNet-21k as unseen ones. For AWA2 and CUB, we utilize the splits proposed by Xian et al. (2017) while randomly assigning aircraft types into 50 seen and 20 unseen classes.

As the classification performance on unseen classes is primarily contingent on the validity and semantic continuity of the class representation, the proposed setup can assess the visual alignment of language embeddings. In the absence of image-specific captions, text-based class representations $\boldsymbol{f}(y_i)$ are used as supervision signals during training and for zero-shot transfer during inference. Besides the template-based targets proposed by Radford et al. (2021) that solely substitute the respective class names, we generate more comprehensive auxiliary information about classes (e.g., visual descriptions) using the instruction-tuned version of the Llama-3 8B model and acquire human-curated information from Wikipedia (details provided in A.2).

## 5    LANGUAGE MODELS FOR VISUAL ZERO-SHOT GENERALIZATION

Utilizing the Visual Text Representation Benchmark(ViTeRB), we investigate the impact of specific design choices to identify critical factors that promote generalization and inform subsequent decisions when building a locked-image-locked-text model.

**LLMs are comprehensive repositories for real-world knowledge.**    While simple templates have proven effective classification targets on large-scale CLIP-like models (Radford et al., 2021; LAION AI, 2022), training with conventional image classification datasets opens up the possibility of using alternative semantic class representations as well. Especially the class-wise supervision combined with limited diversity and number of concepts can impede vision-language alignment. Therefore, we first utilize ViTeRB with different types of textual class representations to gauge how their nature and information content affect the model's generalization ability. More details about the characteristics and acquisition of these class representations are provided in Section A.2 of the appendix.

Table 1: **Classification results of ViTeRB benchmark for various language models.** Decoder-based language models outperform encoder-based architectures across all types of input data. Llama-3 8B is used for LLM generated Wikipedia articles and descriptions.

| Model Type | Language Model | Class Names | LLM Description | LLM Wikip. Articles | Wikip. Articles |
|---|---|---|---|---|---|
| **Encoder** | **BERT-Large** | 18.3 | 15.9 | 20.8 | **22.9** |
| | **T5-XL** | 33.6 | 37.8 | 37.9 | **40.7** |
| | **SentenceT5-XXL** | 39.5 | **44.7** | 44.3 | 41.6 |
| **Decoder** | **NV-Embed** | 40.5 | 42.9 | **47.5** | 45.9 |
| | **Gemma 7B** | 39.7 | 33.7 | **45.1** | 43.4 |
| | **Llama-3 8B** | 40.2 | 43.8 | **44.9** | 44.3 |

Depending on the type of language model, text features are obtained from a special CLS token or the last text token, as shown in Figure 3. The results are summarized in Table 1.

In line with previous studies (Pratt et al., 2022; Menon & Vondrick, 2023; Yang et al., 2022; Saha et al., 2024), we find that the addition of auxiliary information, such as class descriptions, results in improved performance for most language models. This is even true for the Llama-3 model, which was used to generate the description data and resembles findings from Chain-of-thought (Wei et al., 2022), where model performance increases with response length. We also find that LLM-generated articles describing a class in the style of Wikipedia (LLM Wikip Articles in Table A.2 can provide strong targets during multi-modal alignment, achieving the best overall performance of $47.5\%$. Interestingly, relying on strictly human-curated data in the form of actual Wikipedia articles tends to *lower* scores, for example, from $47.5\% \rightarrow 45.9\%$ and $44.9\% \rightarrow 44.3\%$, for NV-Embed and Llama-3. Thus, LLMs can effectively absorb and interpolate substantial amounts of factual information from their training data, positioning them as valuable sources of visually relevant knowledge.

**Decoders outperform encoders in visual concept representation.** A new insight resulting from this analysis is the competitiveness of decoder-based language models for representing visual concepts. Compared to encoders, we show that representing inputs with decoders can result in higher performance for visual tasks, mirroring a recently emerging trend in the language domain (Lee et al., 2024; Springer et al., 2024). NV-Embed (Lee et al., 2024), a model tuned explicitly for embedding text, emerges as the best performer across various types of input data with a maximum performance of $47.5\%$. However, even off-the-shelve LLMs like Gemma or Llama manage to outperform encoder-based models and trail NV-Embed with ViTeRB scores of $45.1\%$ and $44.9\%$, respectively.

**LLM performance correlates with visual performance.** In Figure 1, we compare various LLMs by their ViTeRB performance, as well as their MMLU (Hendrycks et al., 2021) score, which is a common metric to measure LLM performance. We find that the capability of language models is positively correlated with their ability to perform well on the visual ViTeRB tasks. Since models steadily improve in the language domain, this benchmark will be useful to assess whether the trend of increasing visual understanding will continue in future LLM models. If this holds, *ShareLock* can piggyback off and benefit from developments in the LLM domain.

A notable outlier is presented by the Phi-3 model family (Microsoft, 2024), which score comparably low ViTeRB results given their MMLU scores. This discrepancy likely illustrates the effects of the extensive data curation and synthetic data creation utilized in Phi3, which might remove visual information to favor tokens that promote reasoning abilities. Thus, a lack of exposure to sufficient factual knowledge about real-world conditions may impede the formation of visually informed internal representations.

## 6 IMAGE-CAPTION PRETRAINING EXPERIMENTS

The previous section has provided us with prerequisite insights to propose *ShareLock* and motivated the choice to leverage the strong visually aligned representations of LLMs in the context of a CLIP-like model. Forgoing the strict zero-shot setup and moving toward larger-scale image-caption datasets, we intend to explore how well these observations translate in the context of a general-purpose VLM and whether only optimizing a lightweight network on top of frozen features is suf-

ficient to compete with full pretraining or fine-tuning. This analysis will illustrate the current upper limits of utilizing unimodal foundation models as building blocks while applying minimal additional compute and multi-modal data to achieve high-performing VLMs.

## 6.1 EXPERIMENTAL SETUP

**Pretrained Vision and Language Models.** Given its strong performance, broad pretraining regime, and popularity, the ViT-B/14 variant of the DINOv2 model family (Oquab et al., 2023) is used as the default vision backbone unless noted otherwise. Language features are extracted from a Llama-3 8B LLM through last token pooling, as shown in Figure 3. For LiT baselines, we initialize the language encoder with pretrained BERT weights (Devlin et al., 2019), in accordance with the original implementation (Zhai et al., 2022). When comparing LiT, ASIF, and ShareLock models in the following, the exact same pre-computed input features (barring the language component of LiT).

**Projection Networks.** As in Zhai et al. (2022), no transformations are applied to the vision features. The MLP projection network after the language model comprises four layers. Between consecutive layers, inputs are normalized via Batch Normalization (Ioffe & Szegedy, 2015) and fed into a ReLU non-linearity. Dropout (Srivastava et al., 2014) with $p = 0.2$ is applied during training. We have also explored more sophisticated projection networks, but found the MLP to overall provide best performances, see details and ablations in Appendix A.3.

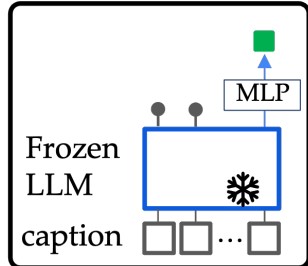

Figure 3: **Text features.** We obtain the final text features by processing the last caption token with an MLP.

**Datasets.** Our investigation focuses on minimizing the amount of paired data required and explores how unimodal embeddings can drive robust multimodal performance with minimal supervision and alignment. As a result, our evaluation is limited to comparably small paired datasets. **COCO Captions.** Containing human annotations for around 80k images, COCO Captions (Chen et al., 2015) is a small but high-quality multimodal dataset. As multiple captions per image are available, a random caption is sampled during each iteration. **CC3M.** The Conceptual Captions dataset (Sharma et al., 2018) was built by scraping image-alt-text pairs from websites, applying filters to remove noisy or mismatched data. Due to expired links, our version of CC3M contains around 2.8M image-text pairs. We also use a smaller subset filtered for more balanced concept coverage for the LLaVA VLM (Liu et al., 2023). **CC12M.** Expanding the scale and diversity of CC3M, CC12M (Changpinyo et al., 2021) is the largest dataset used to train and evaluate our model, offering insights into performance at higher data scales. Our dataset version contains approximately 8.5M image-text pairs.

**Training.** The training setup largely follows the CC12M configuration of LiT (Zhai et al., 2022) and uses the Adam optimizer (Kingma & Ba, 2014) with a learning rate of $10^{-3}$ and a weight decay of $10^{-4}$. Gradient clipping to a global norm of $1$ is applied. The CLIP loss (Radford et al., 2021) with $\tau = 0.07$ is employed, and models are trained until convergence on a validation split sets in, which is around 5k optimization steps with a batch size of $16{,}384$ – regardless of dataset size. Features of the frozen vision and language models are initially precomputed and stored for direct re-use in subsequent epochs.

**Training speed and storage.** On a single A100 48GB GPU, the precomputation of language features with LLama3 8B takes around 8 hours for the CC3M-Llava subset containing 563k image-caption pairs. The DINOv2 features are obtained in 1 GPU hour, and the final multimodal optimization of the MLP also takes around 1 GPU hour. This brings the total training time to around 10 GPU hours. In terms of storage, the original dataset requires around 80GB of storage, while our precomputed features only require around 12GB.

**Evaluation.** We employ a comprehensive suite of VLM evaluations to assess and compare *ShareLock*'s capabilities across a wide range of tasks. Based on the publicly available CLIP Benchmark (LAION AI, 2022), we gauge the models' zero-shot classification and retrieval abilities across diverse datasets and provide qualitative text-to-image retrieval results on ImageNet for CC3M trained

Table 2: **Frozen CLIP-like zero-shot classification on ImageNet variants.** *ShareLock* outperforms CLIP, LiT and ASIF baselines across 21/24 ImageNet evaluations and achieves performances competitive with models that utilize significantly more paired data, such as CommonPool-L (384M).

| Model | Training Dataset | | Test Dataset | | | | | | Average |
|-------|------|------|-------|-------|------|------|-----------|-----------|---------|
| | Size | Name | IN-1k | IN-V2 | IN-R | IN-A | IN Sketch | ObjectNet | |
| LiT | 83k | COCO Captions | 23.3 | 20.8 | 34.4 | 21.1 | 18.4 | 29.2 | 24.5 |
| ASIF | 83k | COCO Captions | 9.4 | 8.7 | 14.4 | 8.8 | 6.9 | 16.1 | 10.7 |
| *ShareLock* | 83k | COCO Captions | **32.2** | **28.6** | **36.6** | **22.8** | **22.4** | **30.4** | **28.8** |
| LiT | 563k | CC3M Subset | 41.7 | 37.5 | 59.2 | 44.4 | 32.4 | 40.7 | 42.6 |
| ASIF | 563k | CC3M Subset | 21.6 | 20.5 | 27.7 | 24.4 | 14.9 | 21.5 | 21.8 |
| *ShareLock* | 563k | CC3M Subset | **50.5** | **45.8** | **60.5** | **47.0** | **36.9** | **41.1** | **47.0** |
| CLIP | 2.8M | CC3M | 16.0 | 13.2 | 17.6 | 3.6 | 6.4 | 8.2 | 10.8 |
| LiT | 2.8M | CC3M | 44.1 | 39.3 | 62.7 | 45.6 | 34.8 | **43.3** | 45.0 |
| *ShareLock* | 2.8M | CC3M | **52.1** | **47.1** | **64.1** | **50.9** | **39.0** | 43.1 | **49.4** |
| DataComp-LAION | 3.84M | CommonPool-S | 3.0 | 2.7 | 4.4 | 1.5 | 1.3 | 3.7 | 2.8 |
| CLIP | 12M | CC12M | 41.6 | 35.4 | 52.6 | 10.7 | 28.8 | 24.0 | 32.2 |
| LiT | 8.5M | CC12M | 56.2 | 49.9 | **70.3** | 52.8 | 43.9 | **47.8** | 53.5 |
| *ShareLock* | 8.5M | CC12M | **59.1** | **53.2** | 68.8 | **53.4** | **44.5** | 46.7 | **54.3** |
| DataComp-LAION | 38.4M | CommonPool-M | 23.0 | 18.9 | 28.0 | 4.3 | 15.1 | 17.7 | 17.8 |
| DataComp-LAION | 384M | CommonPool-L | 55.3 | 47.9 | 65.0 | 20.2 | 43.2 | 46.5 | 46.3 |
| CLIP | 400M | Proprietary | 68.4 | 61.8 | 77.6 | 50.1 | 48.2 | 55.4 | 60.2 |

models. Additionally, the challenging compositionality Winoground task (Thrush et al., 2022) is explored.

**Benchmarks.** We compare our proposed method against a variety of existing VLMs with a particular emphasis on data-efficient alignment approaches. Alongside the original ViT-B/16 variant of CLIP (Radford et al., 2021), we test against several CLIP-like models trained on public datasets of different scales (Fan et al., 2023; Gadre et al., 2023). Using pretrained models to their advantage, we assess how *ShareLock* stacks up against LiT (Zhai et al., 2022) and ASIF (Norelli et al., 2023).

## 6.2 COMPARISON TO STATE-OF-THE-ART

**Comparison to prior works on IN-1k.** Taking the ImageNet-1k zero-shot classification performance as the principal benchmark for model performance, *ShareLock* clearly outperforms other models trained with similar amounts of data, as demonstrated in Table 2. Compared to the small-scale CC3M CLIP model (Fan et al., 2023), *ShareLock* performs notably better, achieving an accuracy 52.1% vs. 16.0%. Adding LLM-based features further proves effective when considering the 44.1% accuracy of LiT (Zhai et al., 2022), which utilizes the same vision backbone as *ShareLock*. Our optimization-based alignment also consistently outperforms the training-free ASIF (Norelli et al., 2023) method, which relies on a large reference dataset with diverse concepts covered for performance. As the dataset size increases, fine-tuning encoders becomes more feasible. Yet, *ShareLock* still maintains performance gains of $3\% - 18\%$ to LiT and CLIP even for CC12M.

**Robustness.** To evaluate the robustness of the VLMs under distribution shifts, the ImageNet-1k classification objective is repeated with visual out-of-distribution inputs. As seen in Table 2, columns 'IN-v2', 'IN-R', etc., *ShareLock* still compares favorably to previous approaches. On average, it surpasses other models trained with datasets comparable in scale and approaches vanilla CLIP models trained with orders of magnitude more data (8.5M vs. 400M for the original CLIP).

**Fine-grained classification.** As shown in Table 3, the strong unimodal features of *ShareLock* similarly contribute positively to fine-grained problems. Here, *ShareLock* outperforms CLIP, LiT, and DataComp models by large margins on 8/12 evaluations. We also find that on these datasets, the models benefit much more noticeably from increased data scale, *e.g.* from 10.6% on Flowers to 48.8% when increasing the dataset $100\times$. Intuitively, exposure to a more diverse and nuanced set of concepts makes methods more capable of performing fine-grained classification. Nonetheless, the effectiveness of our method in leveraging auxiliary knowledge contained in LLM representations is demonstrated through surpassing alternative methods on the same training datasets.

Table 3: **Zero-shot classification on fine-grained datasets.** Fine-grained problems rely heavily on large-scale data; still, *ShareLock* performs competitively with other models trained on the same data.

| Model | Training Dataset | | Test Dataset | | | | | Average |
| | Size | Name | Aircraft | Pets | Flowers | Cars | EuroSAT | |
|---|---|---|---|---|---|---|---|---|
| LiT | 83k | COCO Captions | 1.6 | **28.8** | 7.7 | 1.8 | 21.9 | 12.3 |
| ASIF | 83k | COCO Captions | 2.8 | 7.0 | 1.6 | 1.3 | 21.5 | 6.8 |
| *ShareLock* | 83k | COCO Captions | **3.0** | 20.6 | **10.6** | **9.2** | **25.1** | **13.7** |
| CLIP | 2.8M | CC3M | 1.4 | 13.0 | 10.8 | 0.8 | 12.9 | 7.8 |
| LiT | 2.8M | CC3M | 2.1 | 28.5 | **35.9** | 3.0 | **34.4** | 20.8 |
| *ShareLock* | 2.8M | CC3M | **6.5** | **43.1** | 32.8 | 4.4 | 27.9 | **22.9** |
| DataComp-LAION | 3.84M | CommonPool-S | 1.4 | 4.0 | 1.8 | 1.6 | 15.8 | 4.9 |
| CLIP | 12M | CC12M | 2.5 | 64.2 | 36.7 | **24.1** | 20.9 | 29.7 |
| LiT | 8.5M | CC12M | 5.0 | **74.4** | 48.2 | 13.2 | 35.3 | 35.2 |
| *ShareLock* | 8.5M | CC12M | **8.3** | 66.6 | **48.8** | 11.5 | **40.7** | **36.7** |
| DataComp-LAION | 38.4M | CommonPool-M | 1.7 | 29.9 | 22.4 | 22.0 | 18.8 | 18.9 |
| DataComp-LAION | 384M | CommonPool-L | 7.1 | 77.8 | 53.3 | 67.7 | 41.0 | 49.4 |
| CLIP | 400M | Proprietary | 24.4 | 89.0 | 71.2 | 64.7 | 55.9 | 61.0 |

Table 4: **Compositional reasoning.** Strong frozen language features alone do not address the shortcomings inherent to prior CLIP-like models when it comes to spacial or conceptual relationships. For space, the full table including CC3M model performances is provided in the Appendix.

| Model | Training Dataset | | Winoground | | |
| | Size | Name | Text | Image | Group |
|---|---|---|---|---|---|
| Human | | | **89.5** | 88.5 | 85.5 |
| Chance | | | **25.0** | 25.0 | 16.7 |
| LiT | 83k | COCO Captions | **25.0** | 5.8 | 2.8 |
| ASIF | 83k | COCO Captions | 18.8 | 9.0 | 5.3 |
| *ShareLock* | 83k | COCO Captions | 21.0 | **11.8** | **6.5** |
| CLIP | 12M | CC12M | 22.3 | 9.5 | **5.3** |
| LiT | 8.5M | CC12M | 24.3 | 6.5 | 4.8 |
| *ShareLock* | 8.5M | CC12M | **26.3** | **12.8** | **5.3** |
| DataComp-LAION | 38.4M | CommonPool-M | 25.0 | 8.3 | 6.3 |
| DataComp-LAION | 384M | CommonPool-L | 27.0 | 9.5 | 7.0 |
| CLIP | 400M | Propriatary | 30.8 | 10.8 | 8.3 |

**Compositionality.** Late fusion VLMs have long struggled with nuanced textual scene descriptions or fine-grained compositional differences as tested through benchmarks like Winoground (Thrush et al., 2022) or SugarCrepe (Hsieh et al., 2023). While our results are competitive in Winoground, outscoring CLIP and LiT by a couple of percentage points in 5/6 cases (see Table 4), the reliance on capable pretrained models so far failed to materialize in a significant above-random performance for the challenging compositionality task. *ShareLock* shares similar limitations as previous methods and remains far from reaching human-level performance.

**Data scaling.** In Figure 4, we show the performance of various CLIP-like methods and models with increasing image-caption dataset sizes. We find that starting from scratch, vanilla CLIP models require orders of magnitude more data to achieve similar performance levels compared to *ShareLock*. This remains true even for more sophisticated data filtering methods as used in the DataComp models. While sharing a similar improvement trajectory, suggesting comparable scaling characteristics, *ShareLock* also consistently outperforms LiT. This underlines that features of extensively trained unimodal models possess a semantic understanding that can be efficiently aligned across modalities with minimal paired data.

**Limitations** As shown in the previous section, *ShareLock* outperforms existing methods across zero-shot classification tasks. However, we find that it performs less competitively in other problem settings. As we believe that such negative results can yield useful insights into the workings of our method to the research community, we explicitly highlight some persisting limitations.

For retrieval tasks, having more tunable model components can be advantageous, as seen in Table 5. Here, CLIP and LiT frequently achieve higher scores compared to *ShareLock* with a relative advantage of 10.4 and 1.1 percentage points across evaluation datasets for the CC12M-trained vari-

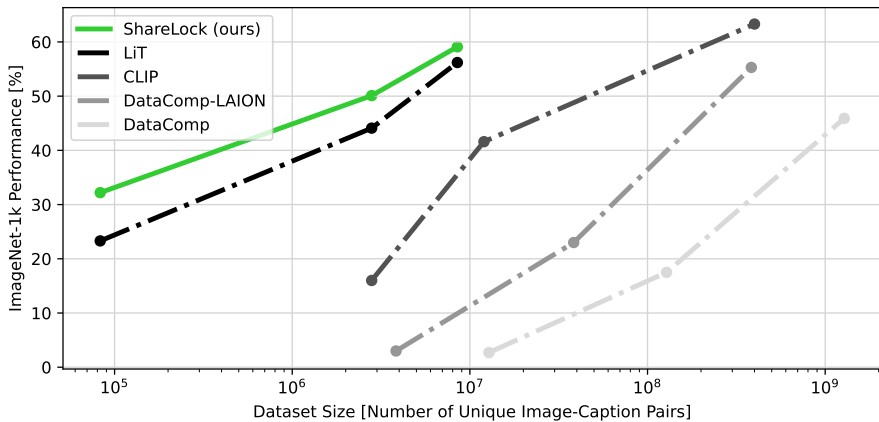

Figure 4: **Scaling of zero-shot performance across dataset sizes.** *ShareLock* achieves best-in-class performance using significantly fewer image-caption pairs compared to CLIP and LiT models.

Table 5: **Recall@5 scores for image and text retrieval.** Previous models with encoder-based and fully fine-tuned language features achieve better performance in most retrieval tasks. For space, the full table including CC3M model performances is provided in the Appendix.

| Model | Training Size | Dataset Name | Flickr8k $\mathcal{T} \to \mathcal{I}$ | Flickr8k $\mathcal{I} \to \mathcal{T}$ | Flickr30k $\mathcal{T} \to \mathcal{I}$ | Flickr30k $\mathcal{I} \to \mathcal{T}$ | MS COCO $\mathcal{T} \to \mathcal{I}$ | MS COCO $\mathcal{I} \to \mathcal{T}$ | Average $\mathcal{T} \to \mathcal{I}$ | Average $\mathcal{I} \to \mathcal{T}$ |
|---|---|---|---|---|---|---|---|---|---|---|
| LiT | 83k | COCO Captions | **57.5** | **77.1** | **61.3** | **80.6** | **50.7** | **69.1** | **56.5** | **75.6** |
| ASIF | 83k | COCO Captions | 10.4 | 20.6 | 12.1 | 24.9 | 8.9 | 17.1 | 10.4 | 20.9 |
| *ShareLock* | 83k | COCO Captions | 50.5 | 69.6 | 56.9 | 78.4 | 27.0 | 41.9 | 50.2 | 69.5 |
| CLIP | 12M | CC12M | **73.1** | **84.5** | **73.9** | **86.3** | **51.0** | **65.4** | **66.0** | **78.7** |
| LiT | 8.5M | CC12M | 60.0 | 72.8 | 69.1 | 82.1 | 36.5 | 53.4 | 55.2 | 69.4 |
| *ShareLock* | 8.5M | CC12M | 55.0 | 73.0 | 67.1 | 81.7 | 32.7 | 50.1 | 51.6 | 68.3 |
| DataComp-LAION | 38.4M | CommonPool-M | 39.4 | 52.6 | 39.2 | 52.4 | 26.0 | 35.9 | 34.9 | 47.0 |
| DataComp-LAION | 384M | CommonPool-L | 78.1 | 89.0 | 81.0 | 90.7 | 57.6 | 71.7 | 72.2 | 83.8 |
| CLIP | 400M | Propriatary | 82.9 | 91.4 | 85.6 | 96.2 | 58.4 | 76.8 | 75.6 | 88.1 |

ants, respectively. This may be due to the reduced internal post-hoc adaptation capacity during contrastive alignment of frozen task-unspecific textual representations. However, when utilizing retrieval-specific LLM features such as from NV-Embed Lee et al. (2024), the *ShareLock* paradigm experiences a significant boost in retrieval abilities of up to $17\%$, as shown in Table 7.

## 6.3 QUALITATIVE RESULTS

In addition to the extensive suite of quantitative evaluations, we present several qualitative results in Figure 5 to illustrate the effectiveness of our method. Across diverse textual prompts, our approach demonstrates strong alignment between textual and visual representations. Compared to versions of CLIP and LiT also trained on CC3M, we find that *ShareLock* generally performs better for fine-grained (*i.e., "a photo of a BMW."*) and more abstract (*i.e., "[...] heavy seas."*) prompts.

## 6.4 ABLATIONS

As the nature of the frozen input features is of great significance in the *ShareLock* paradigm, the choice of vision and language encoders is ablated. In addition, alternative projection network architectures are compared in Section A.3. All ablations are performed on the CC3M dataset.

**Vision encoder.** *ShareLock* is agnostic to the specific type of vision encoder employed. Therefore, we ablate different choices of supervised and unsupervised supervision approaches as well as different model architectures in Table 6. As *ShareLock* projects language embeddings into the vision space, the model choice is pivotal, as demonstrated by the DINOv2 backbone yielding almost double the classification scores compared to the worst performing encoder and a $36\%$ lead over the second-best vision transformer. Models supervised on datasets limited in scope clearly show

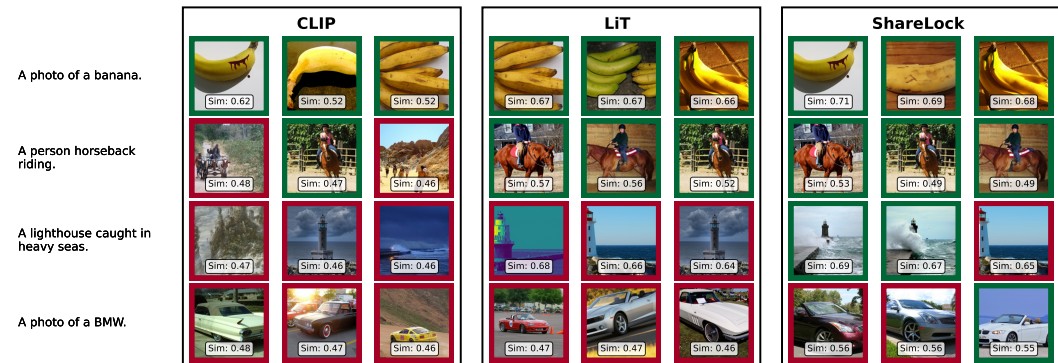

Figure 5: **Comparison on text-to-image retrieval.** We show qualitative top-3 retrieval results for CLIP, LiT and *ShareLock* all trained on CC3M. Green border color indicates correctly retrieved samples.

Table 6: **Ablation of the vision encoder used in *ShareLock*.** Strong and comprehensive image features are essential for generalization.

| Vision Encoder | Dataset | Avg. Classification Scores | | Avg. Retrieval Score | | WinoGround | | |
| | | Robustness | Fine-Grained | $\mathcal{T} \to \mathcal{I}$ | $\mathcal{I} \to \mathcal{T}$ | Text | Image | Group |
| --- | --- | --- | --- | --- | --- | --- | --- | --- |
| ResNet101 | IN-1k (sup.) | 28.0 | 16.5 | 28.3 | 44.9 | 21.8 | 15.3 | 8.8 |
| ViT-B/16 | IN-1k (sup.) | 36.2 | 14.8 | 36.0 | 47.3 | 21.8 | 10.8 | 6.0 |
| DINO ViT-B/16 | IN-1k (unsup.) | 25.8 | 16.1 | 38.8 | 54.5 | 21.5 | 15.3 | 8.8 |
| ***ShareLock* (DINOv2)** | Various (unsup.) | **49.4** | **22.9** | **48.2** | **62.7** | **22.8** | **15.8** | **9.0** |

Table 7: **Ablation of language model used for dual-locked tuning.** Decoder-based language models are key to enable the strong performance of *ShareLock*.

| Language Model | Avg. Classification Scores | | Avg. Retrieval Score | | WinoGround | | |
| | Robustness | Fine-Grained | $\mathcal{T} \to \mathcal{I}$ | $\mathcal{I} \to \mathcal{T}$ | Text | Image | Group |
| --- | --- | --- | --- | --- | --- | --- | --- |
| BERT-Base | 36.2 | 7.2 | 36.6 | 50.3 | 19.5 | 9.0 | 5.0 |
| ***ShareLock* (NV-Embed)** | **50.9** | **25.8** | **56.5** | **69.2** | **27.0** | 14.8 | 8.3 |
| ***ShareLock* default (Llama-3 8B)** | 49.4 | 22.9 | 48.2 | 62.7 | 22.8 | **15.8** | **9.0** |

reduced robustness and less generality compared to DINOv2, illustrating how generalization can benefit from broad pretraining across various concepts – even without explicit supervision.

**Language model.** Similarly, we compare variations of *ShareLock* using different language models and list their performance in Table 7. These results illustrate the effectiveness of decoder-based approaches previously highlighted by the ViTeRB benchmark in Section 5. Despite serving as the starting point in LiT models, the BERT encoder fails to achieve competitive results without any fine-tuning. In contrast, frozen decoder-based representations consistently outperform their BERT-based counterparts, with improvements ranging from 40% to 450%. This demonstrates the expressiveness and high information content of strong LLM representations obtained through text-only pretraining.

# 7 CONCLUSION

We introduce ViTeRB, a benchmark for evaluating the visual capabilities and alignment of language models. With it, we show that LLM quality, measured by MMLU, correlates with visual understanding, and decoder-based LLMs excel in extracting visually informed representations. Building on these insights, we propose *ShareLock*, a simple CLIP-like VLM that leverages the large-scale pretraining and internal knowledge of frozen LLMs. Our method achieves strong performances and requires fewer image-caption pairs than models like CLIP or LiT for similar performances. Combined with its extremely fast training time, this work highlights the potential of frozen decoder-only LLMs for vision-language tasks.

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

## A APPENDIX

### A.1 REPRODUCIBILITY STATEMENT

We acknowledge and emphasize the importance of reproducibility in our work and take active measures to facilitate reproducibility efforts. Besides providing comprehensive documentation of our methods throughout the main paper, with additional details in the supplementary materials, we will publish source code for the proposed *ShareLock* model.

### A.2 TEXTUAL CLASS REPRESENTATIONS FOR VISUAL TEXT REPRESENTATION BENCHMARK

More details about the characteristics and acquisition of these class representations are provided in Section A.2 of the appendix. Besides the template-based targets proposed by Radford et al. (2021) that solely substitute the respective class names, we generate more comprehensive auxiliary information about classes (e.g., visual descriptions) using the instruction-tuned version of the Llama-3-8B model and acquire human-curated information from Wikipedia (details provided in A.2).

Class representations are essential for facilitating the knowledge transfer between classes in the traditional definition of zero-shot learning. Compared to attributes or other forms of class semantics, language-based class representations are more conveniently accessible at various scales and may come in diverse manifestations. The advent of LLMs adds further possibilities for generating and obtaining such auxiliary information. The following paragraphs specify the respective properties and acquisition process. Here, all LLM-based class representations are generated using the instruct-tuned version of LLama-3 8B.

**Class Names.** A set of 80 human-engineered prompt templates in the style of `"a photo of a <class name>"` are adopted from Radford et al. (2021).

**Description.** This type of class representation is generated by tasking an LLM to generate short, one-sentence descriptions of how a given class looks like. Multiple descriptions are generated for each class by slightly varying the LLM prompt and utilizing different seeds as a form of augmentation.

**Wikipedia Page.** Being a comprehensive and mostly factually correct source of information, Wikipedia constitutes an interesting source of auxiliary information in the context of zero-shot classification. To obtain class-article correspondences, class names are automatically matched with page names, after which additional manual quality checks are performed. Nonetheless, an ideal match does not always exist due to high class specificity or generality, in which case superordinate articles are considered or template-based fallbacks are employed.

**LLM-based Wikipedia Style Articles.** Despite being specifically prompted for articles mimicking Wikipedia, the Llama-3-generated texts tend to show significant differences in style compared to their real counterparts.

As the lengthy nature of Wikipedia(-style) articles might dilute the information content captured by the language embeddings, the texts are split into individual sentences, which are used as targets during training. For all types of class representations, predictions are made by aggregating class scores through averaging over all individual class-specific texts.

### A.3 PROJECTION NETWORK ARCHITECTURE

The multi-layer perceptron (MLP) projection networks of *ShareLock* as introduced in Section 3 are conceivably simple. As these are the only unfrozen and tunable parts of the model architecture and thus responsible for aligning vision and language inputs, they are of particular significance to aptly process and transform the inputs. Following Zhai et al. (2022), no transformation to the vision inputs is applied for any of the architectures. With a hidden size of 4096 and four layers, the MLP processing the language features comprises approximately 53M parameters.

In addition to the straightforward MLP-based networks, also more sophisticated Transformer-based architectures are inspired by recent works. First introduced as part of the BLIP-2 model (Li et al., 2023), the Q-Former is a lightweight Transformer-based model that extracts features from an input modality using cross-attention with learnable query tokens. Similarly, albeit introduced in a different context, NV-Embed (Lee et al., 2024) uses a latent attention layer to pool language tokens and receive a global embedding. Slight adjustments are made to both baseline architectures to better suit late-fusion vision-language modeling. The hyperparameters were selected based on the implementation details suggested in the original publications and to approximately match the MLP baseline in learnable parameter count. Both the Q-Former and the NV-Embed projection networks have a token dimension of 1024 in the Transformer parts of the models, eight learnable queries (Q-Former), and key/values (NV-Embed). Whereas the Q-Former consists of 3 blocks and 4 attention heads, NV-Embed comprises a total of four layers with eight cross-attention heads each.

The choice of projection network architecture is ablated in Table 8. While no single architecture consistently scores best, the MLP-based *ShareLock* configuration performs competitively compared to NV-Embed and QFormer throughout the evaluation cases. Additionally, Transformer-based architectures entail increased computational complexity due to the more evolved attention mechanism and processing of more tokens, making MLPs an attractive choice from an efficiency perspective as well. These results suggest that the additional information contained across all tokens of an input is not significantly more adjuvant compared to solely considering the last token representation as is done with the MLP.

Table 8: **Ablation of the projection network architectures tuned as part of *ShareLock* training.** Simple MLPs perform competitively compared to more advanced Transformer-based architectures.

| Architecture | Avg. Classification Scores | | Avg. Retrieval Score | | WinoGround | | |
|---|---|---|---|---|---|---|---|
| | Robustness | Fine-Grained | $\mathcal{T} \to \mathcal{I}$ | $\mathcal{I} \to \mathcal{T}$ | Text | Image | Group |
| NV-Embed | 41.9 | 20.3 | **49.5** | **65.8** | 21.5 | 10.0 | 6.0 |
| QFormer | 48.3 | **26.8** | 49.4 | 65.2 | **24.5** | 14.0 | 8.5 |
| *ShareLock* (MLP) | **49.4** | 22.9 | 48.2 | 62.7 | 22.8 | **15.8** | **9.0** |

## A.4 SUPPLEMENTARY QUANTITATIVE RESULTS

The following tables include additional results and analyses that were omitted in the main body of the paper due to space constraints. These supplementary results offer extended insights from additional model variants and further buttress previously drawn conclusions.

Table 9: **Extended results for zero-shot classification on ImageNet variants.**

| Model | Training Dataset | | Test Dataset | | | | | | Average |
|---|---|---|---|---|---|---|---|---|---|
| | Size | Name | IN-1k | IN-V2 | IN-R | IN-A | IN Sketch | ObjectNet | |
| LiT | 83k | COCO Captions | 23.3 | 20.8 | 34.4 | 21.1 | 18.4 | 29.2 | 24.5 |
| ASIF | 83k | COCO Captions | 9.4 | 8.7 | 14.4 | 8.8 | 6.9 | 16.1 | 10.7 |
| *ShareLock* | 83k | COCO Captions | **32.2** | **28.6** | **36.6** | **22.8** | **22.4** | **30.4** | **28.8** |
| LiT | 563k | CC3M Subset | 41.7 | 37.5 | 59.2 | 44.4 | 32.4 | 40.7 | 42.6 |
| ASIF | 563k | CC3M Subset | 21.6 | 20.5 | 27.7 | 24.4 | 14.9 | 21.5 | 21.8 |
| *ShareLock* | 563k | CC3M Subset | **50.5** | **45.8** | **60.5** | **47.0** | **36.9** | **41.1** | **47.0** |
| CLIP | 2.8M | CC3M | 16.0 | 13.2 | 17.6 | 3.6 | 6.4 | 8.2 | 10.8 |
| LiT | 2.8M | CC3M | 44.1 | 39.3 | 62.7 | 45.6 | 34.8 | **43.3** | 45.0 |
| *ShareLock* | 2.8M | CC3M | **52.1** | **47.1** | **64.1** | **50.9** | **39.0** | 43.1 | **49.4** |
| DataComp-LAION | 3.84M | CommonPool-S | 3.0 | 2.7 | 4.4 | 1.5 | 1.3 | 3.7 | 2.8 |
| CLIP | 12M | CC12M | 41.6 | 35.4 | 52.6 | 10.7 | 28.8 | 24.0 | 32.2 |
| LiT | 8.5M | CC12M | 56.2 | 49.9 | **70.3** | 52.8 | 43.9 | **47.8** | 53.5 |
| *ShareLock* | 8.5M | CC12M | **59.1** | **53.2** | 68.8 | **53.4** | **44.5** | 46.7 | **54.3** |
| DataComp | 12.8M | CommonPool-S | 2.7 | 2.3 | 4.1 | 1.4 | 1.1 | 3.7 | 2.5 |
| DataComp | 128M | CommonPool-M | 17.5 | 14.4 | 19.8 | 3.9 | 9.5 | 15.8 | 13.5 |
| DataComp-LAION | 38.4M | CommonPool-M | 23.0 | 18.9 | 28.0 | 4.3 | 15.1 | 17.7 | 17.8 |
| DataComp-LAION | 384M | CommonPool-L | 55.3 | 47.9 | 65.0 | 20.2 | 43.2 | 46.5 | 46.3 |
| CLIP | 400M | Proprietary | 68.4 | 61.8 | 77.6 | 50.1 | 48.2 | 55.4 | 60.2 |
| DataComp | 1.28B | CommonPool-L | 45.9 | 39.2 | 52.7 | 15.9 | 34.5 | 41.1 | 38.2 |
| DataComp-LAION | 3.84B | CommonPool-XL | 75.4 | 68.5 | 87.0 | 57.0 | 63.5 | 68.5 | 70.0 |
| DataComp | 12.8B | CommonPool-XL | 72.3 | 65.1 | 85.9 | 56.4 | 61.1 | 70.6 | 68.6 |

Table 10: **Extended results for zero-shot classification on fine-grained datasets.**

| Model | Training Dataset Size | Name | Test Dataset Aircraft | Pets | Flowers | Cars | EuroSAT | Average |
|---|---|---|---|---|---|---|---|---|
| LiT | 83k | COCO Captions | 1.6 | **28.8** | 7.7 | 1.8 | 21.9 | 12.3 |
| ASIF | 83k | COCO Captions | 2.8 | 7.0 | 1.6 | 1.3 | 21.5 | 6.8 |
| *ShareLock* | 83k | COCO Captions | 3.0 | 20.6 | **10.6** | **9.2** | **25.1** | **13.7** |
| LiT | 563k | CC3M Subset | 1.1 | 22.8 | 27.5 | 4.1 | 25.5 | 16.2 |
| ASIF | 563k | CC3M Subset | 2.1 | 11.7 | 6.4 | 2.3 | 19.5 | 8.4 |
| *ShareLock* | 563k | CC3M Subset | **8.4** | **38.3** | **33.3** | 5.4 | **29.4** | **23.0** |
| CLIP | 2.8M | CC3M | 1.4 | 13.0 | 10.8 | 0.8 | 12.9 | 7.8 |
| LiT | 2.8M | CC3M | 2.1 | 28.5 | **35.9** | 3.0 | **34.4** | 20.8 |
| *ShareLock* | 2.8M | CC3M | 6.5 | 43.1 | 32.8 | 4.4 | 27.9 | 22.9 |
| DataComp-LAION | 3.84M | CommonPool-S | 1.4 | 4.0 | 1.8 | 1.6 | 15.8 | 4.9 |
| CLIP | 12M | CC12M | 2.5 | 64.2 | 36.7 | **24.1** | 20.9 | 29.7 |
| LiT | 8.5M | CC12M | 5.0 | **74.4** | 48.2 | 13.2 | 35.3 | 35.2 |
| *ShareLock* | 8.5M | CC12M | 8.3 | 66.6 | 48.8 | 11.5 | 40.7 | 36.7 |
| DataComp | 12.8M | CommonPool-S | 0.8 | 4.4 | 2.3 | 1.4 | 14.9 | 4.8 |
| DataComp-LAION | 38.4M | CommonPool-M | 1.7 | 29.9 | 22.4 | 22.0 | 18.8 | 18.9 |
| DataComp | 128M | CommonPool-M | 1.3 | 16.8 | 8.1 | 13.3 | 25.4 | 13.0 |
| DataComp-LAION | 384M | CommonPool-L | 7.1 | 77.8 | 53.3 | 67.7 | 41.0 | 49.4 |
| CLIP | 400M | Proprietary | 24.4 | 89.0 | 71.2 | 64.7 | 55.9 | 61.0 |
| DataComp | 1.28B | CommonPool-L | 3.3 | 56.2 | 39.4 | 60.5 | 33.4 | 38.6 |
| DataComp-LAION | 3.84B | CommonPool-XL | 93.1 | 77.1 | 28.7 | 89.2 | 73.8 | 72.4 |
| DataComp | 12.8B | CommonPool-XL | 19.5 | 90.6 | 71.6 | 89.3 | 68.9 | 68.0 |

Table 11: **Extended results for image and text retrieval.**

| Model | Training Dataset Size | Name | Flickr8k $\mathcal{T} \to \mathcal{I}$ | $\mathcal{I} \to \mathcal{T}$ | Flickr30k $\mathcal{T} \to \mathcal{I}$ | $\mathcal{I} \to \mathcal{T}$ | MS COCO $\mathcal{T} \to \mathcal{I}$ | $\mathcal{I} \to \mathcal{T}$ | Average $\mathcal{T} \to \mathcal{I}$ | $\mathcal{I} \to \mathcal{T}$ |
|---|---|---|---|---|---|---|---|---|---|---|
| LiT | 83k | COCO Captions | **57.5** | **77.1** | **61.3** | **80.6** | **50.7** | **69.1** | **56.5** | **75.6** |
| ASIF | 83k | COCO Captions | 10.4 | 20.6 | 12.1 | 24.9 | 8.9 | 17.1 | 10.4 | 20.9 |
| *ShareLock* | 83k | COCO Captions | 50.5 | 69.6 | 56.9 | 78.4 | 27.0 | 41.9 | 50.2 | 69.5 |
| LiT | 563k | CC3M Subset | **51.2** | **67.2** | **60.7** | **74.4** | 28.4 | 45.8 | **46.8** | **62.5** |
| ASIF | 563k | CC3M Subset | 10.3 | 20.3 | 15.9 | 29.6 | 5.7 | 12.1 | 10.6 | 20.7 |
| *ShareLock* | 563k | CC3M Subset | 49.9 | 64.6 | 57.9 | 73.9 | **29.8** | **45.9** | 44.9 | 60.1 |
| CLIP | 2.8M | CC3M | 43.5 | 56.9 | 40.4 | 54.7 | 25.3 | 30.9 | 36.4 | 47.5 |
| LiT | 2.8M | CC3M | **60.1** | **76.6** | **69.3** | **81.4** | 35.9 | 53.6 | **55.1** | **70.5** |
| *ShareLock* | 2.8M | CC3M | 54.9 | 70.1 | 60.1 | 74.2 | 29.5 | 43.9 | 48.2 | 62.7 |
| DataComp-LAION | 3.84M | CommonPool-S | 7.8 | 11.4 | 6.7 | 9.1 | 3.6 | 4.4 | 6.1 | 8.3 |
| CLIP | 12M | CC12M | **73.1** | **84.5** | **73.9** | **86.3** | 51.0 | 65.4 | **66.0** | **78.7** |
| LiT | 8.5M | CC12M | 60.0 | 72.8 | 69.1 | 82.1 | 36.5 | 53.4 | 55.2 | 69.4 |
| *ShareLock* | 8.5M | CC12M | 55.0 | 73.0 | 67.1 | 81.7 | 32.7 | 50.1 | 51.6 | 68.3 |
| DataComp | 12.8M | CommonPool-S | 8.1 | 12.3 | 6.9 | 9.9 | 3.5 | 5.7 | 6.2 | 9.3 |
| DataComp-LAION | 38.4M | CommonPool-M | 39.4 | 52.6 | 39.2 | 52.4 | 26.0 | 35.9 | 34.9 | 47.0 |
| DataComp | 128M | CommonPool-M | 30.7 | 42.3 | 31.4 | 40.7 | 19.4 | 30.0 | 27.2 | 37.7 |
| DataComp-LAION | 384M | CommonPool-L | 78.1 | 89.0 | 81.0 | 90.7 | 57.6 | 71.7 | 72.2 | 83.8 |
| CLIP | 400M | Propriatary | 82.9 | 91.4 | 85.6 | 96.2 | 58.4 | 76.8 | 75.6 | 88.1 |
| DataComp | 1.28B | CommonPool-L | 64.3 | 78.6 | 69.9 | 81.4 | 45.7 | 60.2 | 60.0 | 73.4 |
| DataComp-LAION | 3.84B | CommonPool-XL | 90.9 | 96.1 | 92.9 | 99.0 | 71.4 | 84.6 | 85.1 | 93.2 |
| DataComp | 12.8B | CommonPool-XL | 84.6 | 92.1 | 86.4 | 94.6 | 63.1 | 77.1 | 78.0 | 87.9 |

## A.5 SUPPLEMENTARY QUALITATIVE RESULTS

Figure 6 provides additional qualitative insights into the retrieval ability of CLIP, LiT, and ShareLock models trained on CC3M.

Table 12: **Extended results for compositional reasoning.**

| Model | Training Dataset Size | Name | Winoground Text | Image | Group | SugarCrepe Replace | Swap | Add |
|---|---|---|---|---|---|---|---|---|
| Human | | | **89.5** | 88.5 | 85.5 | **99.6** | **99.5** | **99.0** |
| Chance | | | **25.0** | 25.0 | 16.7 | **50.0** | **50.0** | **50.0** |
| LiT | 83k | COCO Captions | **25.0** | 5.8 | 2.8 | **78.7** | **65.1** | **75.7** |
| ASIF | 83k | COCO Captions | 18.8 | 9.0 | 5.3 | 49.1 | 44.3 | 46.8 |
| *ShareLock* | 83k | COCO Captions | 21.0 | **11.8** | **6.5** | 70.5 | 55.4 | 68.4 |
| LiT | 563k | CC3M Subset | **24.3** | 8.3 | 5.5 | 69.5 | 57.7 | 67.2 |
| ASIF | 563k | CC3M Subset | 18.3 | 13.3 | **7.3** | 58.7 | 52.1 | 56.8 |
| *ShareLock* | 563k | CC3M Subset | 20.0 | **13.8** | 6.8 | 62.4 | 50.6 | 60.8 |
| CLIP | 2.8M | CC3M | 21.3 | 9.5 | 6.0 | 67.0 | 56.6 | 63.3 |
| LiT | 2.8M | CC3M | **23.8** | 6.0 | 4.5 | **74.0** | **62.4** | **73.6** |
| *ShareLock* | 2.8M | CC3M | 22.8 | **15.8** | **9.0** | 63.0 | 54.0 | 62.3 |
| DataComp-LAION | 3.84M | CommonPool-S | 19.3 | 12.0 | 7.5 | 56.5 | 53.6 | 58.1 |
| CLIP | 12M | CC12M | 22.3 | 9.5 | **5.3** | 77.5 | 61.9 | 73.5 |
| LiT | 8.5M | CC12M | 24.3 | 6.5 | 4.8 | **74.1** | **62.0** | **77.6** |
| *ShareLock* | 8.5M | CC12M | **26.3** | 12.8 | **5.3** | 66.3 | 53.1 | 65.5 |
| **DataComp** | 12.8M | CommonPool-S | 17.3 | 5.5 | 2.3 | 57.7 | 51.7 | 56.4 |
| DataComp-LAION | 38.4M | CommonPool-M | 25.0 | 8.3 | 6.3 | 69.1 | 56.7 | 66.2 |
| **DataComp** | 128M | CommonPool-M | 24.3 | 4.5 | 3.0 | 65.5 | 53.4 | 65.5 |
| DataComp-LAION | 384M | CommonPool-L | 27.0 | 9.5 | 7.0 | 79.8 | 62.8 | 79.3 |
| CLIP | 400M | Propriatary | 30.8 | 10.8 | 8.3 | 80.0 | 62.7 | 73.0 |
| **DataComp** | 1.28B | CommonPool-L | 24.0 | 6.5 | 4.3 | 73.4 | 58.7 | 75.2 |
| **DataComp-LAION** | 3.84B | CommonPool-XL | 34.0 | 11.8 | 10.0 | 79.7 | 58.7 | 81.4 |
| **DataComp** | 12.8B | CommonPool-XL | 28.8 | 7.5 | 6.0 | 84.3 | 66.7 | 87.5 |

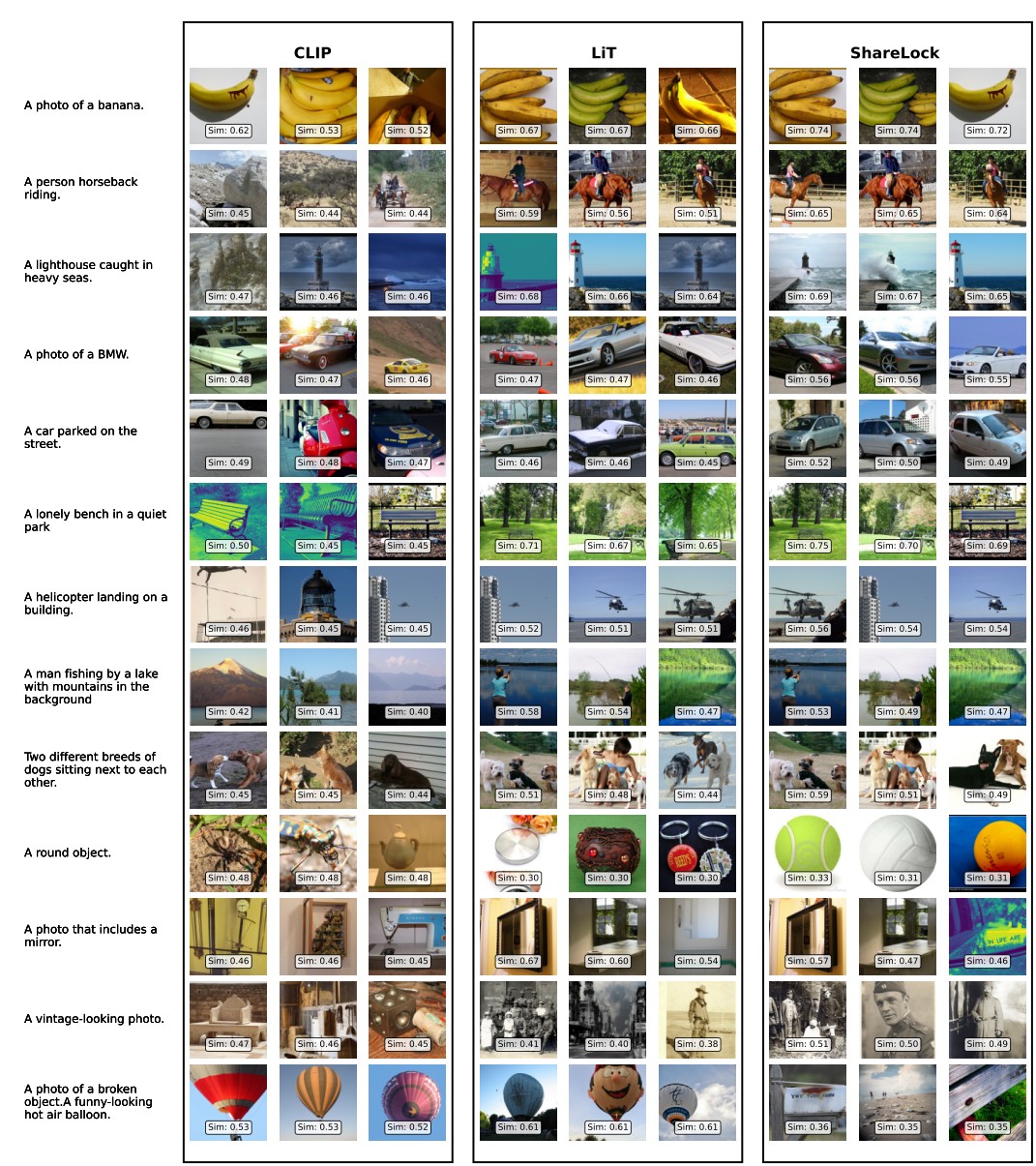

Figure 6: **Qualitative comparison on text-to-image retrieval (ImageNet-1k).**

