# OpenReview forum: "Do better language models have crisper vision?"
_ICLR.cc/2025/Conference — ICLR 2025 Conference Withdrawn Submission_

### Official Review · Reviewer_dAG5 · 2024-10-30

**Soundness:** 3
**Presentation:** 2
**Contribution:** 3
**Rating:** 6
**Confidence:** 4

**Summary:**

This paper introduced ViTeRB, a benchmark for evaluating the visual capabilities and alignment of language models. It shown that LLM quality correlates with visual understanding ability, and decoder-only language models excel in extracting visually informed representations. Further, a CLIP-like VLM training method, ShareLock, is proposed to leverage the large-scale pretraining and internal knowledge of frozen LLMs, which frozen both image and text models.

**Strengths:**

-The introduced ViTeRB benchmark is interesting and can evaluate the visual capabilities and alignment of language models.

-The proposed VLM training is very efficient, since both image and text models are frozen. The performance is high compared to previous methods.

**Weaknesses:**

-It is still unclear for me that how to perform the zero-shot classification on image datasets. Since only MLP projection layers for text are tuned, we cannot perform image classification. It is mentioned in line 197, "text-based class representation f(y) are used as supervision signals during training and for zero-shot transfer during inference...". Can you give a more explicit explanation on it? Also, Table 2 and Table 3 are zero-shot classification results. Are they have the same setting as in Table 1?

-A better zero-shot classification performance is good. However, the ShareLock cannot help tune the image encoder, thus facilitating the visual understanding with numerous knowledge. Also, the performance on VQA is more important than zero-shot classification in LLM era. Can you provide more results regarding to building the LMM?

**Questions:**

See the weaknesses.

---

### Official Review · Reviewer_aRbq · 2024-11-04

**Soundness:** 1
**Presentation:** 3
**Contribution:** 1
**Rating:** 3
**Confidence:** 3

**Summary:**

This paper investigates the visual understanding ability of large language models. The authors proposed the visual text representation benchmark to evaluate the visual understanding ability of LLMs. Plus, the authors develop ShareLock for efficient and effective CLIP-style model tunning with pre-trained LLMs and vision models.

**Strengths:**

- The paper is well-organized.

- The motivation for analysis of the visual understanding ability is interesting.

**Weaknesses:**

- The contribution of the paper needs further justification.

- The evaluation method for the visual understanding ability needs further discussion.

**Questions:**

- 1. The proposed ShareLock architecture is similar to CLIP-Adapter[1], which adds an MLP to the top of the Image model. In this work, the authors add an MLP on the top of the text model. The reviewer thinks there is no big difference between these two manners.
    - The authors do not discuss the difference between ShareLock and CLIP-Adapter. The reviewer thinks such discussions are necessary.

- 2. For the evaluation results in Table 1, the reviewer thinks the results cannot represent the visual understanding ability of LLMs.
    - 2.1. In Section 4, the authors say "It retains the model architecture and optimization objectives of ShareLock". In such case, the reviewer thinks results in Table 1 are obtained by tunning ShareLock with an image encoder and text encoder. This cannot present the visual understanding ability of LLMs. It is just a reflection of the alignment results of image features and text features.
    - 2.2. The dividing of seen and unseen classes just tests the zero-shot ability of the tuned ShareLock, not the image model and text model.
    - 2.3. The relative changes of metrics in Table 1 may only reflect which LLMs have features that better align with the image model, not their visual understanding ability. For example, LLM A is good at visual understanding but poorly aligns with the image model, LLM B is relatively bad at visual understanding but well aligns with the image model.
    - 2.4. In summary, **the reviewer does not think the ViTeRB benchmark accurately reflects the visual understanding ability of LLMs.**

- 3. In Table 2, where should we find the results of LiT in the original LiT paper[2]? From the results in [2], the review thinks ShareLock is not better than LiT. For example, in Table 1 of LiT, it achieves 75.7% zero-shot transfer accuracy on ImageNet.
    - The reviewer thinks the data in the paper (Table 2&3) are all reproduced by the authors. The reviewer thinks some third-party baselines/results should be introduced to make the results more convincible.


[1] https://arxiv.org/pdf/2110.04544

[2] https://arxiv.org/pdf/2111.07991

---

### Official Review · Reviewer_1Sy7 · 2024-11-04

**Soundness:** 3
**Presentation:** 3
**Contribution:** 2
**Rating:** 6
**Confidence:** 4

**Summary:**

The authors propose ShareLock, a CLIP-like model that utilizes frozen LLMs (LLaMA 8B), frozen vision models (DINOv2), and a lightweight trainable MLP layer. ShareLock achieves competitive results in zero-shot image classification with minimal data and compute resources. ShareLock demonstrates state-of-the-art efficiency, reaching 51% accuracy on ImageNet with only 563k image-caption pairs and a total training time of 10 GPU hours. This work highlights the potential of decoder-based LLMs as the text branch in CLIP-style VL models, offering an efficient and scalable approach to multimodal learning.

**Strengths:**

1. ViTeRB provides a structured means to evaluate the visual knowledge contained within language models, addressing a gap in assessing their visual understanding.
1. ShareLock uses frozen LLM and vision features, leading to substantial efficiency gains, making it feasible to achieve strong performance with minimal data and compute resources.
1. ShareLock outperforms existing methods on ImageNet classification with far fewer image-text pairs and training resources, highlighting the potential for data-efficient VLM training.
1. The study finds that decoder-based LLMs are more effective in encoding visual representations compared to encoder-based textual models, which could inspire future vision-language model designs.
1. ShareLock scales smoothly with more image-text paired data as shown in Fig4.

**Weaknesses:**

1. The conclusion of "Decoders outperform encoders in visual concept representation." drawn in Sec 5 of this paper may be questionable. This paper poses it as if it's the model architecture difference that causes this performance difference. However, the comparison is unfair. For example, the number of model parameters is different, the amount of data these language models were trained on is different. Furthermore, the style of data these language models were trained on is different so the encoder-based model may not be able to properly leverage self-generated class descriptions. The reviewer suggests revising the writing of this part to avoid potentially misleading reasons and conclusions.

1. (Optional) It would add value to this paper to also show how much data is required so that ShareLock can surpass the performance of a CLIP model of the same size.

**Questions:**

1. The authors demonstrated the scaling of ShareLock with respect to the amount of data. What about the scaling with respect to the model sizes? For example, in CLIP, as the ViT scales up, the ImageNet zero-shot performance also becomes better. Does ShareLock also have this property as we scale up either the ViT or the LLM?

1. ShareLock lags behind other methods in retrieval, and the authors suspect that it's because of the limited adaptation capability of frozen models. Please provide more analysis of this part or prove this claim by, for example, unfreezing the model.

---

### Official Review · Reviewer_bxQW · 2024-11-08

**Soundness:** 3
**Presentation:** 3
**Contribution:** 2
**Rating:** 5
**Confidence:** 4

**Summary:**

This paper proposed a method called ShareLock, which connects a frozen vision encoder and a frozen language decoder with an MLP projector on the embeddings produced by a language decoder, for classification and retrieval tasks. This method is shown to be significantly better than LiT and ASIF, when trained on small scale image-text pair datasets. Thanks to the frozen image and language models, ShareLock is also quite compute efficient with precomputed embeddings.

**Strengths:**

* Novel idea of using language decoders as text feature extractor to pair with frozen vision encoders just with a small MLP projector. This paper shows that decoders outperforms encoders for visual concept representation.
* This paper shows correlation between language tasks and its vision transfer capabilities (Figure 1).
* The original LiT paper shows that freezing the language tower doesn’t work well for classification or retrieval tasks. This paper addressed this problem by using embeddings from decoders with additional MLP projectors, and showed good results.

**Weaknesses:**

* The contribution of Visual Text Representation Benchmark (ViTeRB) is a bit unclear. Comparison of this new benchmark with existing benchmarks would be able to justify it. For example, does ViTeRB capture different information? Would ViTeRB rank methods differently compared with the existing standard benchmarks? If so, it would be nice to report rank correlation between the new benchmark and the existing benchmarks.
* Though strong results are obtained with COCO Captions, CC3M or CC12M, it would be nice to also present results with larger datasets from DataComp or any other sources. As highlighted in the abstract, the training time is only 1 GPU hour (+ 8 GPU hours for precomputing), it would be more convincing to show results on large scale datasets thanks to the reduced GPU time requirements.
* Although using decoders for CLIP-like classification or retrieval tasks are not very common, using decoders for VLM tasks like captioning and question answering are very common (LLAVA, BLIP2, PaliGemma, etc.). This may limit the impact of this paper given the experimental results on the CLIP-like tasks are also pretty small scale.

**Questions:**

* Have you tried different vision encoders other than DINOv2?
* I was hope to see much better results with the proposed method on compositionality tasks (Winoground or SugarCrepe), due to a proper language model (instead of a contrastively pre-trained short text encoder) being used to extract features. Do you know why the results fall into the same range as (and slightly better than) the baselines (CLIP, LIT, ASIF, etc.)?

---

### Author Response · Authors · 2024-11-14

We thank the reviewers for their time and comments. We have decided to incorporate these and further improvements in a future revised submission.

---

### Note · Authors · 2024-11-14

I have read and agree with the venue's withdrawal policy on behalf of myself and my co-authors.